# GRAPH-RELATIONAL DOMAIN ADAPTATION

**Zihao Xu[1]\*, Hao He[2], Guang-He Lee[2], Yuyang Wang[3], Hao Wang[1]**
[1]Rutgers University, [2]Massachusetts Institute of Technology, [3]AWS AI Labs
`zihao.xu@rutgers.edu, {haohe, guanghe}@mit.edu,`
`yuyawang@amazon.com, hw488@cs.rutgers.edu`

## ABSTRACT

Existing domain adaptation methods tend to treat every domain equally and align them all perfectly. Such uniform alignment ignores topological structures among different domains; therefore it may be beneficial for nearby domains, but not necessarily for distant domains. In this work, we relax such uniform alignment by using a domain graph to encode domain adjacency, e.g., a graph of states in the US with each state as a domain and each edge indicating adjacency, thereby allowing domains to align flexibly based on the graph structure. We generalize the existing adversarial learning framework with a novel graph discriminator using encoding-conditioned graph embeddings. Theoretical analysis shows that at equilibrium, our method recovers classic domain adaptation when the graph is a clique, and achieves non-trivial alignment for other types of graphs. Empirical results show that our approach successfully generalizes uniform alignment, naturally incorporates domain information represented by graphs, and improves upon existing domain adaptation methods on both synthetic and real-world datasets[1].

## 1 INTRODUCTION

Generalization of machine learning methods hinges on the assumption that training and test data follows the same distribution. Such an assumption no longer holds when one trains a model in some domains (source domains), and tests it in other domains (target domains) where data follows different distributions. Domain adaptation (DA) aims at improving performance in this setting by aligning data from the source and target domains so that a model trained in source domains can generalize better in target domains (Ben-David et al., 2010; Ganin et al., 2016; Tzeng et al., 2017; Zhang et al., 2019).

Existing DA methods tend to enforce uniform alignment, i.e., to treat every domain equally and align them all perfectly. However, in practice the domains are often heterogeneous; one can expect DA to work well when the source domains are close to the target domains, but not when they are too far from each other (Zhao et al., 2019; Wang et al., 2020). Such heterogeneity can often be captured by a *graph*, where the domains realize the nodes, and the adjacency between two domains can be captured by an edge (see Fig. 1). For example, to capture the similarity of weather in the US, we can construct a graph where each state is treated as a node and the physical proximity between two states results in an edge. There are also many other scenarios where the relation among domains can be naturally captured by a graph, such as the taxonomies of products in retail or connections among research fields of academic papers. Given a domain graph, we can tailor the adaptation of domains to the graph, rather than dictating the data from all the domains to align perfectly regardless of the graph structure.

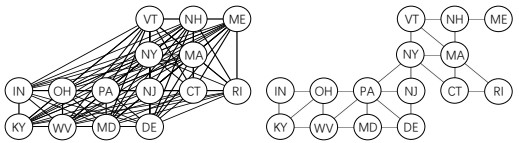

Figure 1: 15 states in the east of the US as 15 domains. **Left:** Traditional DA treats each domain equally and enforces uniform alignment for all domains, which is equivalent to enforcing a fully connected domain graph. **Right:** Our method generalizes traditional DA to align domains according to any specific domain graph, e.g., a domain graph describing adjacency among these 15 states.

---

\*Work conducted during internship at AWS AI Labs.

[1]Code will soon be available at `https://github.com/Wang-ML-Lab/GRDA`

One naïve DA method for such graph-relational domains is to perform DA for each pair of neighboring domains separately. Unfortunately, due to the strict alignment between each domain pair, this method will still lead to uniform alignment so long as the graph is connected. To generalize DA to the graph-relational domains, we argue that an ideal method should (1) only enforce uniform alignment when the domain graph is a clique (i.e., every two domains are adjacent), and (2) more importantly, relax uniform alignment to adapt more flexibly across domains according to any non-clique domain graph, thereby naturally incorporating information on the domain adjacency. In this paper, we generalize adversarial DA methods and replace the traditional binary (or multi-class) discriminator with a novel graph discriminator: instead of distinguishing among different domains, our graph discriminator takes as input the encodings of data to reconstruct the domain graph. We show that our method enjoys the following theoretical guarantees: it recovers classic DA when the the domain graph is a clique, and realizes intuitive alignments for other types of graphs such as chains and stars (see Fig. 4). We summarize our contributions as follows:

- We propose to use a graph to characterize domain relations and develop graph-relational domain adaptation (GRDA) as the first general adversarial DA method to adapt across domains living on a graph.
- We provide theoretical analysis showing that at equilibrium, our method can retain the capability of uniform alignment when the domain graph is a clique, and achieve non-trivial alignment for other types of graphs.
- Empirical results on both synthetic and real-world datasets demonstrate the superiority of our method over the state-of-the-art DA methods.

## 2  RELATED WORK

**Adversarial Domain Adaptation.** There have been extensive prior works on domain adaptation (Pan & Yang, 2009; Pan et al., 2010; Long et al., 2018; Saito et al., 2018; Sankaranarayanan et al., 2018; Zhang et al., 2019; Peng et al., 2019; Chen et al., 2019; Dai et al., 2019; Nguyen-Meidine et al., 2021). Typically they aim to align the distributions of the source and target domains with the hope that the predictor trained on labeled source data can generalize well on target data. Such alignment can be achieved by either directly matching their distributions' statistics (Pan et al., 2010; Tzeng et al., 2014; Sun & Saenko, 2016; Peng et al., 2019; Nguyen-Meidine et al., 2021) or training deep learning models with an additional adversarial loss (Ganin et al., 2016; Zhao et al., 2017; Tzeng et al., 2017; Zhang et al., 2019; Kuroki et al., 2019; Chen et al., 2019; Dai et al., 2019). The latter, i.e., adversarial domain adaptation, has received increasing attention and popularity because of its theoretical guarantees (Goodfellow et al., 2014; Zhao et al., 2018; Zhang et al., 2019; Zhao et al., 2019), its ability to train end-to-end with neural networks, and consequently its promising empirical performance. Also loosely related to our work is continuously indexed domain adaptation (CIDA) Wang et al. (2020), which considers domains indexed by continuous values. These methods typically treat every domain equally and enforce uniform alignment between source-domain data and target-domain data; this is done by generating domain-invariant encodings, where domain invariance is achieved by training the encoder to fool the discriminator that *classifies the domain index*. In contrast, we naturally relax such uniform alignment using a graph discriminator to *reconstruct a domain graph* that describes domain adjacency.

**Domain Adaptation Related to Graphs.** There are also works related to both DA and graphs. Usually they focus on adaptation between two domains where data points themselves are graphs. For example, (Pilancı & Vural, 2019; Pilanci & Vural, 2020) use frequency analysis to align the data graphs between the source domain and the target domains, and (Alam et al., 2018; Ding et al., 2018) perform label propagation on the data graph.

In contrast, GRDA considers a setting completely different from the above references. Instead of focusing on adapting between two domains with data points in the form of graphs (e.g., *each data point itself is a node*), GRDA adapts across multiple domains (e.g., with each state in the US as a domain) according to a domain graph (i.e., *each domain is a node*). Therefore the methods above are not applicable to our setting. Note that (Mancini et al., 2019) uses metadata-weighted batch normalization to propagate information among domains with similar metadata, but it is not an adversarial domain adaptation method. It is orthogonal to GRDA and can be used as a backbone network to further improve GRDA's performance (see the Appendix for empirical results). It is also

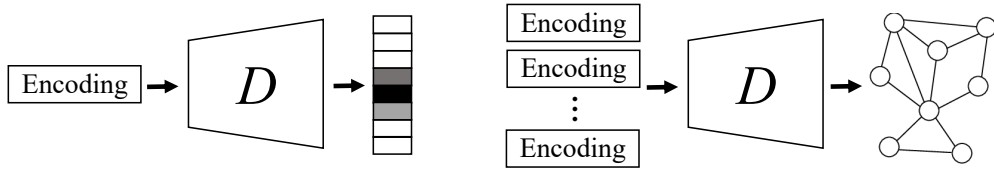

Classifying the Domain Index                    Reconstructing the Domain Graph

Figure 2: Difference between discriminators in traditional DA methods and the graph discriminator in GRDA. **Left:** In traditional DA methods, the discriminator classifies the domain index given an encoding. **Right:** In GRDA, the graph discriminator reconstructs the domain graph given encodings of data from different domains.

worth noting that, in this paper, we assume the domain graph is given. It would be interesting future work to combine GRDA with domain relation inference methods (e.g., domain embeddings (Peng et al., 2020)) when there is a natural but unobserved graph relation among domains.

## 3   METHOD

In this section, we will first briefly introduce the problem setting and then elaborate our domain adaptation method.

### 3.1   PROBLEM SETTING AND NOTATION

We focus on the unsupervised domain adaptation setting with $N$ domains in total. Each domain has a discrete domain index $u \in \mathcal{U} = [N] \triangleq \{1, \ldots, N\}$, belonging to either the source domain index set $\mathcal{U}_s$ or the target domain index set $\mathcal{U}_t$. The relationship between domains is described by a domain graph with the adjacency matrix $\mathbf{A} = [\mathbf{A}_{ij}]$, where $i$ and $j$ index nodes (domains) in the graph. Given labeled data $\{(\mathbf{x}_l^s, y_l^s, u_l^s)\}_{l=1}^n$ from source domains ($u_l^s \in \mathcal{U}_s$), unlabeled data $\{\mathbf{x}_l^t, u_l^t\}_{l=1}^m$ from target domains ($u_l^t \in \mathcal{U}_t$), and the domain graph described by $\mathbf{A}$, we want to predict the label $\{y_l^t\}_{l=1}^m$ for data from target domains. Note that the domain graph is defined on domains with each domain (node) containing multiple data points.

### 3.2   GRAPH-RELATIONAL DOMAIN ADAPTATION (GRDA)

**Overview.** We use an adversarial learning framework to perform adaptation across graph-relational domains. The adversarial game consists of three players: (1) an encoder $E$, which takes as input a datum $\mathbf{x}_l$, the associated domain index $u_l$, and the adjacency matrix $\mathbf{A}$ to generate an encoding $\mathbf{e}_l = E(\mathbf{x}_l, u_l, \mathbf{A})$, (2) a predictor $F$, which makes predictions based on the encoding $\mathbf{e}_l$, and (3) a graph discriminator $D$, which guides the encoding to adapt across graph-relational domains. Specifically, the discriminator $D$ takes in a mini-batch of $B$ encodings $\mathbf{e}_l$ ($l \in [B]$), and tries to reconstruct the domain graph $\mathbf{A}$. By letting the encoder $E$ play adversarially against the discriminator $D$, the graph-relational information of domains will be removed from the encoding $\mathbf{e}_l$ in order to make the discriminator $D$ incapable of reconstructing the graph. Note that the graph discriminator in our adversarial game is different from classic discriminators which classify the domain index, as shown in Fig. 2.

Formally, GRDA performs a minimax optimization with the following loss function:

$$\min_{E,F} \max_D L_f(E, F) - \lambda_d L_d(D, E), \tag{1}$$

where $L_f(E, F)$ is the predictor loss and $L_d(D, E)$ is the discriminator loss, and $\lambda_d$ is a hyperparameter balancing them two. Below we discussed these two terms in detail.

**Predictor.** In Eqn. 1, the predictor loss $L_f(E, F)$ is defined as

$$L_f(E, F) \triangleq \mathbb{E}^s[h_p(F(E(\mathbf{x}_l, u_l, \mathbf{A})), y)],$$

where the expectation $\mathbb{E}^s$ is taken over the source-domain data distribution $p^s(\mathbf{x}, y, u)$. $h_p(\cdot, \cdot)$ is a predictor loss function for the task (e.g., $L_2$ loss for regression).

**Encoder and Node Embeddings.** Given an input tuple $(\mathbf{x}_l, u_l, \mathbf{A})$, the encoder $E$ first computes a graph-informed domain embedding $\mathbf{z}_{u_l}$ based on the domain index $u_l$ and the domain graph $\mathbf{A}$. Then we feed $\mathbf{z}_{u_l}$ along with $\mathbf{x}_l$ into a neural network to obtain the final encoding $\mathbf{e}_l$. Formally we have

$$\mathbf{e}_l = E(\mathbf{x}_l, u_l, \mathbf{A}) = f(\mathbf{x}_l, \mathbf{z}_{u_l}), \tag{2}$$

where $f(\cdot, \cdot)$ is a trainable neural network.

In theory, any embeddings for node (domain) indices should work equally well so long as they are distinct from one another (thus forming a bijection to the set of domains $[N]$). Here we pre-train the embeddings by a reconstruction loss for simplicity, and our intuition is that good embeddings of nodes should inform us of (thus reconstruct) the graph structure. Suppose the nodes indices $i$ and $j$ are sampled independently and identically from the marginal domain index distribution $p(u)$; the reconstruction loss is written as

$$L_g = \mathbb{E}_{i,j \sim p(u)}[-\mathbf{A}_{ij} \log \sigma(\mathbf{z}_i^\top \mathbf{z}_j) - (1 - \mathbf{A}_{ij}) \log(1 - \sigma(\mathbf{z}_i^\top \mathbf{z}_j))],$$

where $\sigma(x) = \frac{1}{1+e^{-x}}$ is the sigmoid function. Note that in general we could use any node embedding methods (Grover & Leskovec, 2016; Tang et al., 2015; Kipf & Welling, 2016b), but this is not the focus of this paper. For fair comparison, we use exactly the same encoder, i.e., $E(\mathbf{x}, u, \mathbf{A})$, for all the methods in the experiments of Sec. 5.

**Graph Discriminator.** The discriminator loss $L_d(D, E)$ in Eqn. 1 is defined as

$$L_d(D, E) \triangleq \mathbb{E}_{(\mathbf{x}_1, u_1),(\mathbf{x}_2, u_2)}[h(\mathbf{x}_1, u_1, \mathbf{x}_2, u_2)], \tag{3}$$

$$h(\mathbf{x}_1, u_1, \mathbf{x}_2, u_2) = -\mathbf{A}_{u_1, u_2} \log \sigma(\widehat{\mathbf{z}}_1^\top \widehat{\mathbf{z}}_2) - (1 - \mathbf{A}_{u_1, u_2}) \log(1 - \sigma(\widehat{\mathbf{z}}_1^\top \widehat{\mathbf{z}}_2)),$$

where $\widehat{\mathbf{z}}_1 = D(E(\mathbf{x}_1, u_1, \mathbf{A}))$, $\widehat{\mathbf{z}}_2 = D(E(\mathbf{x}_2, u_2, \mathbf{A}))$ are the discriminator's reconstructions of node embeddings. The expectation $\mathbb{E}$ is taken over a pair of i.i.d. samples $(\mathbf{x}_1, u_1)$,$(\mathbf{x}_2, u_2)$ from the joint data distribution $p(\mathbf{x}, u)$. The discriminator loss $L_d(D, E)$ essentially quantifies how well the reconstructed node embedding $\widehat{\mathbf{z}}_1, \widehat{\mathbf{z}}_2$ preserve the information of the original connections, or, equivalently, $\mathbf{A}$. We refer readers to the Appendix for detailed model architectures.

Due to the adversarial nature of how the discriminator $D$ and the encoder $E$ engage with the loss, the discriminator $D$ would aim to recover the domain graph via the adjacency structure ($\mathbf{A}$), while the encoder $E$ would prevent the discriminator $D$ from doing so. Intuitively, if the discriminator $D$ is powerful enough to uncover any information regarding the domain graph in the encoding $\mathbf{e}_l$, the optimal encoder $E$ will have to remove all the information regarding graph-relational domains in the encoding $\mathbf{e}_l$, thus successfully adapting across graph-relational domains. We will formally elaborate the above arguments further in the next section.

# 4 THEORY

In this section, we first provide theoretical guarantees that at equilibrium GRDA aligns domains according to any given domain graph. Specifically, we analyze a game where the graph discriminator $D$ tries to reconstruct the domain graph while the (joint) encoder $E$ tries to prevent such reconstruction. To gain more insight on the encoding alignment, we then further discuss the implication of these general theoretical results on GRDA's global optimum using simple example graphs such as cliques, stars, and chains. **All proofs of lemma, theorem and corollary can be found in Appendix A.**

## 4.1 ANALYSIS FOR GRDA

For simplicity, we first consider a game which only involves the joint encoder $E$ and the graph discriminator $D$ with extension to the full game later. Specifically, we focus on following loss function:

$$\max_E \min_D L_d(D, E), \tag{4}$$

where $L_d(D, E)$ is defined in Eqn. 3.

Lemma 4.1 below analyzes the optimal graph discriminator $D$ given a fixed joint encoder $E$. Intuitively, it states that given two encodings $\mathbf{e}$ and $\mathbf{e}'$, the optimal $D$ will output the *conditional expectation* of $\mathbf{A}_{ij}$ over all possible combinations of domain pairs $(i, j)$ sampled from $p(u|\mathbf{e})$ and $p(u|\mathbf{e}')$.

**Lemma 4.1** (**Optimal Discriminator for GRDA**). *For E fixed, the optimal D satisfies following equation,*

$$\sigma(D(\mathbf{e})^\top D(\mathbf{e}')) = \mathbb{E}_{i \sim p(u|\mathbf{e}), j \sim p(u|\mathbf{e}')}[\mathbf{A}_{ij}],$$

*where* $\mathbf{e}$ *and* $\mathbf{e}'$ *are from the encoding space.*

**Theorem 4.1** (**Global Optimum for GRDA with General Domain Graphs**). *Given an adjacency matrix* $\mathbf{A}$*, the total loss in Eqn. 4 has a tight upper bound:*

$$L_d(D, E) \leq H(\mathbb{E}_{\mathbf{e}, \mathbf{e}'} \alpha(\mathbf{e}, \mathbf{e}')) = H(\mathbb{E}_{i,j}[\mathbf{A}_{ij}]),$$

*where* $H$ *denotes the entropy function,* $H(p) = -p \log(p) - (1-p) \log(1-p)$*, and* $\alpha(\mathbf{e}, \mathbf{e}') = \mathbb{E}_{i \sim p(u|\mathbf{e}), j \sim p(u|\mathbf{e}')}[\mathbf{A}_{ij}]$*. Furthermore, the equality, i.e., the optimum, is achieved when*

$$\alpha(\mathbf{e}, \mathbf{e}') = \mathbb{E}_{i,j}[\mathbf{A}_{ij}], \text{ for any } e, e',$$

*or equivalently,* $\mathbb{E}_{i,j}[\mathbf{A}_{ij}|\mathbf{e}, \mathbf{e}'] = \mathbb{E}_{i,j}[\mathbf{A}_{ij}]$*, where* $(\mathbf{e}, \mathbf{e}')$ *is from the encoding space.*

Essentially, Theorem 4.1 shows that if trained to the global optimum, GRDA aligns the domain distribution $p(u|\mathbf{e})$ for all locations in the embedding space such that $\alpha(\mathbf{e}_i, \mathbf{e}_j)$ is identical to the *constant*, $\mathbb{E}_{ij}[\mathbf{A}_{ij}]$, for any pair $(\mathbf{e}_i, \mathbf{e}_j)$.

We can directly apply Theorem 4.1 to evaluate the training process. During training, if the discriminator loss finally converges to the theoretical optimum $H(\mathbb{E}_{i,j}[\mathbf{A}_{ij}])$ (Figure 3), we would know that our training succeeds.

An interesting property of GRDA is that uniform alignment is a solution for any domain graph, as shown in Corollary 4.1 below.

**Corollary 4.1.** *For GRDA, the global optimum of total loss* $L_d(D, E)$ *is achieved if the encoding of all domains (indexed by* $u$*) are perfectly (uniformly) aligned, i.e.,* $\mathbf{e} \perp u$*.*

Note that replacing the inner product inside $\sigma(\cdot)$ with other functions (e.g., $L_2$ distance) does not change Lemma 4.1 and Theorem 4.1, as long as cross-entropy loss is used to consider both positive and negative edges in the graph.

We can also analyze the full game as defined in Eqn. 1 (see the Appendix for details).

**Theorem 4.2.** *If the encoder* $E$*, the predictor* $F$ *and the discriminator* $D$ *have enough capacity and are trained to reach optimum, any global optimal encoder* $E^*$ *has the following properties:*

$$H(y|E^*(\mathbf{x}, u, \mathbf{A})) = H(y|\mathbf{x}, u, \mathbf{A}), \quad (5)$$

*where* $H(\cdot|\cdot)$ *denotes the conditional entropy.*

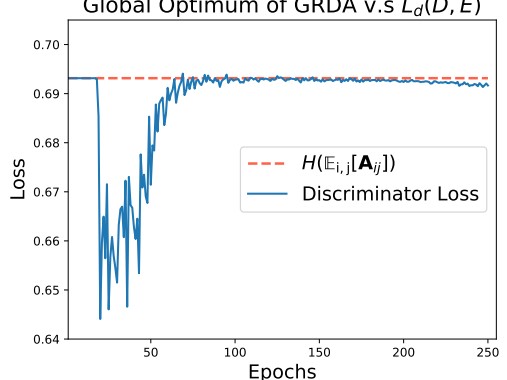

Figure 3: Global Optimum of GRDA v.s. $L_d(D, E)$ during training process. The loss of the discriminator finally converges to the theoretical optimum $H(\mathbb{E}_{i,j}[\mathbf{A}_{ij}])$.

Theorem 4.2 implies that, at equilibrium, the optimal joint encoder $E$ produces encodings that preserve all the information about the label $y$ contained in the data $\mathbf{x}$ and the domain index $u$. Therefore, the global optimum of the two-player game between $E$ and $D$ matches the global optimum of the three-play game between $E$, $D$, and $F$.

## 4.2 IMPLICATION OF GRDA'S GLOBAL OPTIMUM

Sec. 4.1 analyzes the global optimum of GRDA in its general form. In this section, we further discuss the implication of these general theoretical results using simple example graphs such as cliques, stars, and chains; we show that GRDA's global optimum actually translates to domain-graph-informed constraints on the embedding distributions of every domain.

With $N$ domains in total, we assume each domain contains the same amount of data, i.e. $p(u = 1) = \cdots = p(u = N) = N^{-1}$. We further use $p_i(\mathbf{e})$ as a shorthand for $p(\mathbf{e}|u = i)$, the embedding

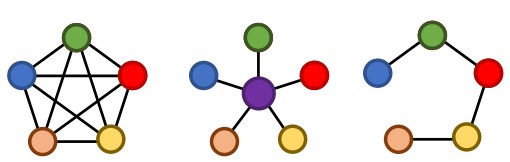
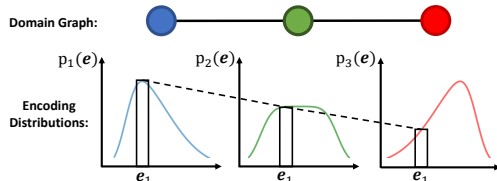

Figure 4: Example domain graphs as discussed in Sec. 4.2. **Left:** Cliques. **Middle:** Star graphs. **Right:** Chain graphs.

Figure 5: Possible encoding distributions of three domains forming a chain graph of three nodes at equilibrium. We can see that for any encoding, e.g., $\mathbf{e}_1$, we have $p_2(\mathbf{e}_1) = \frac{1}{2}(p_1(\mathbf{e}_1) + p_3(\mathbf{e}_1))$.

distribution in domain $i$. With our assumption, the marginal embedding distribution is the average embedding distribution over all domains, i.e., $p(\mathbf{e}) = N^{-1} \sum_{i=1}^{N} p_i(\mathbf{e})$. We define $\beta_i(\mathbf{e}) = p_i(\mathbf{e})/p(\mathbf{e})$, as the ratio between domain-specific embedding distribution and the marginal embedding distribution.

**Cliques (Fully Connected Graphs).** In this example, the $N$ domains in our domain graph are fully connected, i.e. there is an edge between any pair of domains, as shown in Fig. 4 (left).

**Corollary 4.2** (**Cliques**). *In a clique, the GRDA optimum is achieved if and only if the embedding distributions of all the domains are the same, i.e. $p_1(\mathbf{e}) = \cdots = p_N(\mathbf{e}) = p(\mathbf{e}), \forall \mathbf{e}$.*

Interestingly, Corollary 4.2 shows that at equilibrium, GRDA recovers the uniform alignment of classic domain adaptation methods when the domain graph is a clique. Among all connected graphs with $N$ nodes (domains), a clique has the largest number of edges, and therefore GRDA enforces the least flexible alignment, i.e., uniform alignment, when the domain graph is a clique.

**Star Graphs.** In this example, we have $N$ domains forming a star graph (as shown in Fig. 4 (middle)), where domain 1 is the center of the star while all other domains are connected to domain 1.

**Corollary 4.3** (**Star Graphs**). *In a star graph, the GRDA optimum is achieved if and only if the embedding distribution of the center domain is the average of all peripheral domains, i.e. $p_1(\mathbf{e}) = \frac{1}{N-1} \sum_{i=2}^{N} p_i(\mathbf{e}), \forall \mathbf{e}$.*

Corollary 4.3 shows that compared to cliques, GRDA enforces a much more flexible alignment when the domain graph is a star graph. It only requires the embedding distribution of the center domain to be the average of all peripheral domains. In this case, peripheral domains can have very different embedding distributions.

**Chain Graphs.** In this example, we consider the case where $N$ domains form a chain, i.e., domain $i$ is only connected to domain $i-1$ (unless $i = 1$) and domain $i+1$ (unless $i = N$), as shown in Fig. 4 (right).

**Corollary 4.4** (**Chain Graphs**). *Let $p_i(\mathbf{e}) = p(\mathbf{e}|u = i)$ and the average encoding distribution $p(\mathbf{e}) = N^{-1} \sum_{i=1}^{N} p(\mathbf{e}|u = i)$. In a chain graph, the GRDA optimum is achieved if and only if $\forall \mathbf{e}, \mathbf{e}'$*

$$\sum_{i=1}^{N-1} \frac{p_i(\mathbf{e})p_{i+1}(\mathbf{e}') + p_i(\mathbf{e}')p_{i+1}(\mathbf{e})}{p(\mathbf{e})p(\mathbf{e}')} = 2(N-1).$$

Corollary 4.4 shows that in a chain graph of $N$ domains, direct interactions only exist in domain pairs$(i, i-1)$ (i.e., consecutive domains in the chain).

**Chain of Three Nodes.** To gain more insight, below we analyze a special case of chain graphs with three domains, i.e., a chain graph of three nodes as the domain graph, where domain 2 is connected to both domain 1 and domain 3. Note that this is also a special case of star graphs, with one center node and two peripheral nodes. By Corollary 4.3 or Corollary 4.4, we can prove the following Proposition 4.1.

**Proposition 4.1** (**Chain of Three Nodes**). *In this length three chain graph, the GRDA optimum is achieved if and only if the embedding distribution of the middle domain is the average of the embedding distributions of the domains on the two sides, i.e. $p_2(\mathbf{e}) = \frac{1}{2}(p_1(\mathbf{e}) + p_3(\mathbf{e})), \forall \mathbf{e}$.*

Proposition 4.1 states that, for a chain graph of 3 domains, GRDA relaxes classical DA requiring perfect alignment of embedding distributions, and instead only requires linear interpolation/extrapolation

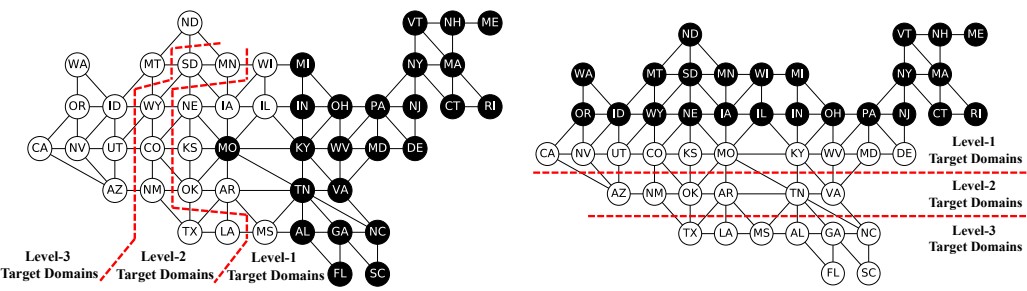

Figure 6: Domain graphs for the two adaptation tasks on *TPT-48*, with black nodes indicating source domains and white nodes indicating target domains. **Left:** Adaptation from the 24 states in the east to the 24 states in the west. **Right:** Adaptation from the 24 states in the north to the 24 states in the south.

relations among embedding distributions. Fig. 5 shows a simple example. If domain 2 (the center domain) is the target domain while others are source domains, GRDA interpolates the embedding distribution of domain 2 according to domain 1 and 3. Similarly, if domain 3 (the peripheral domain on the right) is the target domain while others are source domains, GRDA extrapolates the embedding distribution of domain 3 according to domain 1 and 2.

The theorems above show that GRDA can enforce different levels of alignment, from perfect alignment to linearly interpolating/extrapolating embedding distributions, according to the domain graph.

## 5 EXPERIMENTS

In this section, we compare GRDA with existing methods on both synthetic and real-world datasets.

### 5.1 DATASETS

**DG-15.** We make a synthetic binary classification dataset with 15 domains, which we call *DG-15*. Each domain contains 100 data points. To ensure adjacent domains have similar decision boundaries, we generate *DG-15* as follows. In each domain $i$, we first randomly generate a 2-dimensional unit vector $[a_i, b_i]$ as the node embedding for each domain $i$ and denote the angle of the unit vector as $\omega_i = \arcsin(\frac{b_i}{a_i})$. We then randomly generate positive data $(\mathbf{x}, 1, i)$ and negative data $(\mathbf{x}, 0, i)$ from two different 2-dimensional Gaussian distributions, $\mathcal{N}(\boldsymbol{\mu}_{i,1}, \mathbf{I})$ and $\mathcal{N}(\boldsymbol{\mu}_{i,0}, \mathbf{I})$, respectively, where $\boldsymbol{\mu}_{i,1} = [\frac{\omega_i}{\pi}a_i, \frac{\omega_i}{\pi}b_i]$ and $\boldsymbol{\mu}_{i,0} = [-\frac{\omega_i}{\pi}a_i, -\frac{\omega_i}{\pi}b_i]$. To construct the domain graph, we sample $\mathbf{A}_{ij} \sim Bern(0.5a_ia_j + 0.5b_ib_j + 0.5)$, where $Bern(\theta)$ is a Bernoulli distribution with the parameter $\theta$. With such a generation process, domains with similar node embeddings $[a_i, b_i]$ will (1) have similar $\boldsymbol{\mu}_{i,1}$ and $\boldsymbol{\mu}_{i,0}$, indicating similar decision boundaries, and (2) are more likely to have an edge in the domain graph. We select 6 connected domains as the source domains and use others as target domains, as shown in Fig. 7.

**DG-60.** We make another synthetic dataset with the same procedure as *DG-15*, except that it contains 60 domains, with 6,000 data points in total. The dataset is called *DG-60*. We again select 6 connected domains as the source domains, with others as target domains.

**TPT-48.** *TPT-48* contains the monthly average temperature for the 48 contiguous states in the US from 2008 to 2019. The raw data are from the National Oceanic and Atmospheric Administration's Climate Divisional Database (nClimDiv) and Gridded 5km GHCN-Daily Temperature and Precipitation Dataset (nClimGrid) (Vose et al., 2014). We use the data processed by Washington Post (WP, 2020).

Here we focus on the regression task to predict the next 6 months' temperature based on previous first 6 months' temperature. We consider two DA tasks as illustrated in Fig. 6:

- $E\,(24) \to W\,(24)$: Adapting models from the 24 states in the east to the 24 states in the west.

- $N\,(24) \to S\,(24)$: Adapting models from the 24 states in the north to the 24 states in the south.

We define target domains one hop away from the closest source domain as *Level-1 Target Domains*, those two hops away as *Level-2 Target Domains*, and those more than two hops away as *Level-3 Target Domains*, as shown in Fig. 6.

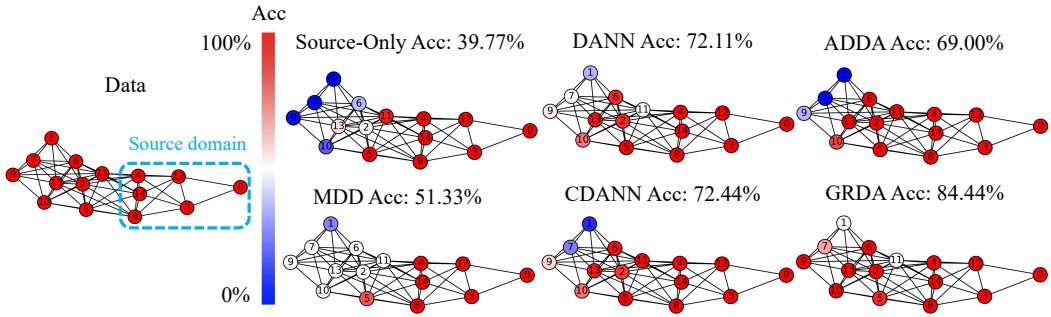

Figure 7: Detailed results on *DG-15* with 15 domains. On the left is the domain graph for *DG-15*. We use the 6 domains in the dashed box as source domains. On the right are the accuracy of various DA methods for each domain, where the spectrum from 'red' to 'blue' indicates accuracy from 100% to 0% (best viewed in color).

Table 1: Accuracy (%) on *DG-15* and *DG-60*.

| Method | Source-Only | DANN | ADDA | CDANN | MDD | GRDA (Ours) |
|--------|-------------|------|------|-------|-----|-------------|
| *DG-15* | 39.77 | 72.11 | 69.00 | 72.44 | 51.33 | **84.44** |
| *DG-60* | 38.39 | 61.98 | 32.17 | 61.70 | 66.24 | **96.76** |

**CompCars.** *Comprehensive Cars* (CompCars) (Yang et al., 2015) contains 136,726 images of cars with labels including car types, viewpoints, and years of manufacture (YOMs). Our task is to predict the car type based on the image, with each view point and each YOM as a seperate domain. We choose a subset of CompCars with 4 car types (MPV, SUV, sedan and hatchback), 5 viewpoints (front (F), rear (R), side (S), front-side (FS), and rear-side (RS)), ranging from 2009 to 2014. It has 30 domains (5 viewpoints × 6 YOMs) and 24,151 images in all. We choose cars with front view and being produced in 2009 as a source domain, and the others as targets domains. We consider two domains connected if either their viewpoints or YOMs are identical/nearby. For example, Domain A and B are connected if their viewpoints are 'F' and 'FS', respectively.

## 5.2 BASELINES AND IMPLEMENTATION

We compared our proposed GRDA with state-of-the-art DA methods, including Domain Adversarial Neural Networks (**DANN**) (Ganin et al., 2016), Adversarial Discriminative Domain Adaptation (**ADDA**) (Tzeng et al., 2017), Conditional Domain Adaptation Neural Networks (**CDANN**) (Zhao et al., 2017), and Margin Disparity Discrepancy (**MDD**) (Zhang et al., 2019). We also report results when the model trained in the source domains is directly tested in the target domains (**Source-Only**). Besides the baselines above, we formulated the DA task as a semi-supervised learning (SSL) problem and adapted two variants of graph neural networks (Kipf & Welling, 2016a; Zhou et al., 2020). We prove that the first variant cannot work (Theorem C.1), and our experiments show that the second variant performs worse than most domain adaptation baselines. For completeness, we also report the result of SENTRY, a recent entropy-based method, on our tasks (see the Appendix for details). All the algorithms are implemented in PyTorch (Paszke et al., 2019) and the balancing hyperparameter $\lambda_d$ is chosen from 0.1 to 1 (see the Appendix for more details on training). Note that since MDD is originally designed for classification tasks, we replace its cross-entropy loss with an $L_2$ loss so that it could work in the regression tasks on *TPT-48*. For fair comparison, all the baselines use **x**, the domain index $u$, and the node embedding **z** as inputs to the encoder (as mentioned in Eqn. 2).

## 5.3 RESULTS

***DG-15* and *DG-60*.** We show the accuracy of the compared methods in Table 10. Based on the table, it is clear that training on the source domains without adaptation (Source-Only) leads to even worse results than random guess (50% accuracy), simply due to overfitting the source domains under domain shifts. While existing DA methods such as DANN and MDD may outperform Source-Only in most of the cases, their improvements over random guess are limited. In some cases, they barely improve upon 'random prediction' (e.g., MDD on *DG-15*) or even underperform Source-Only (e.g., ADDA

Table 2: MSE for various DA methods for both tasks E (24) → W (24) and N (24) → S (24) on *TPT-48*. We report the average MSE of all domains as well as more detailed average MSE of Level-1, Level-2, Level-3 target domains, respectively (see Fig. 6). Note that there is only one single DA model per column. We mark the best result with **bold face** and the second best results with underline.

| Task | Domain | Source-Only | DANN | ADDA | CDANN | MDD | GRDA (Ours) |
|------|--------|-------------|------|------|-------|-----|-------------|
| E (24)→W (24) | Average of 8 Level-1 Domains | 1.852 | 3.942 | 2.449 | **0.902** | 0.993 | 0.936 |
| | Average of 7 Level-2 Domains | 1.992 | 2.186 | 2.401 | 2.335 | 1.383 | **1.025** |
| | Average of 9 Level-3 Domains | 1.942 | 1.939 | 1.966 | 3.002 | **1.603** | 1.830 |
| | Average of All 24 Domains | 1.926 | 2.679 | 2.254 | 2.108 | 1.335 | **1.297** |
| N (24)→S (24) | Average of 10 Level-1 Domains | 2.255 | 1.851 | 2.172 | 0.924 | 1.159 | **0.754** |
| | Average of 6 Level-2 Domains | 1.900 | 1.964 | 2.600 | 6.434 | 2.087 | **0.882** |
| | Average of 8 Level-3 Domains | 3.032 | 4.015 | 3.462 | **1.882** | 3.149 | 1.889 |
| | Average of All 24 Domains | 2.426 | 2.600 | 2.709 | 2.621 | 2.054 | **1.165** |

Table 3: Accuracy (%) on CompCars (4-Way Classification).

| Method | Source-Only | DANN | ADDA | CDANN | MDD | GRDA (Ours) |
|--------|-------------|------|------|-------|-----|-------------|
| *CompCars* | 46.5 | 50.2 | 46.1 | 48.2 | 49.0 | **51.0** |

on *DG-60*). In contrast, GRDA can successfully perform adaptation by aligning data according to the domain graph and achieve much higher accuracy.

Fig. 7 shows the accuracy of different DA methods on *DG-15* for each domain, where the spectrum from 'red' to 'blue' indicates accuracy from 100% to 0%. Without domain adaptation, Source-Only only achieves high accuracy for the domains immediately adjacent to the source domains, while its accuracy drops dramatically when the target domain is far away from the source domains, achieving nearly 0% accuracy in the three most distant domains. While existing DA methods may improve the overall accuracy, all of them are worse than random guess (50% accuracy) in at least one domain. Notably, MDD achieves accuracy of only 50% in almost all the target domains. In contrast, the accuracy of GRDA never falls below 50% for every domain.

***TPT-48.*** Table 2 shows the mean square error (MSE) in both adaptation tasks for different DA methods on the dataset *TPT-48*. In terms of average performance of all the target domains, we can see that DANN, ADDA and CDANN achieve negative performance boost in both $E$ (24) → $W$ (24) and $N$ (24) → $S$ (24), highlighting the difficulty of performing DA across graph-relational domains. MDD can achieve stable improvement upon Source-Only, and our proposed GRDA can further improve the performance, achieving the lowest average MSE in both tasks.

Besides average MSE of all the target domains, we also report more fine-grained results to see how the error distributes across different target domains. Specifically, Table 2 shows the average MSE of Level-1, Level-2, and Level-3 target domains (see Fig. 6 for an illustration) for both tasks. These results show GRDA can consistently achieve low MSE across all the levels of target domains.

***CompCars.*** Table 3 shows the average classification accuracy on CompCars for different DA methods. We found that all the DA methods, except ADDA, outperforms Source Only. Our proposed method GRDA takes full advantage of the domain graph and achieves the most significant accuracy improvement (9.68%) (see the Appendix for more detailed results).

## 6 CONCLUSION

In this paper, we identify the problem of adaptation across graph-relational domains, i.e., domains with adjacency described by a domain graph, and propose a general DA method to address such a problem. We further provide theoretical analysis showing that our method recover uniform alignment in classic DA methods, and achieve non-trivial alignment for other types of graphs, thereby naturally incorporating domain information represented by domain graphs. Empirical results demonstrate our method's superiority over state-of-the-art methods. Future work could include fusing automatic graph discovery with our method, as well as extending our method to multiple or heterogeneous domain graphs with real-value weights. It would also be interesting to consider other applications such as recommender systems and natural language processing.

ACKNOWLEDGEMENT

The authors thank the reviewers/AC for the constructive comments to improve the paper. HW is partially supported by NSF Grant IIS-2127918. The views and conclusions contained herein are those of the authors and should not be interpreted as necessarily representing the official policies, either expressed or implied, of the sponsors.

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

# A  PROOF

**Lemma 4.1** (**Optimal Discriminator for GRDA**). *For E fixed, the optimal D satisfies following equation,*

$$\sigma(D(\mathbf{e})^\top D(\mathbf{e}')) = \mathbb{E}_{i\sim p(u|\mathbf{e}), j\sim p(u|\mathbf{e}')}[\mathbf{A}_{ij}],$$

*where $\mathbf{e}$ and $\mathbf{e}'$ are from the encoding space.*

*Proof.* With $E$ fixed, the optimal $D$ is

$$\underset{D}{\mathrm{argmin}}\, \mathbb{E}_{(\mathbf{x},j),(\mathbf{x}',j)\sim p(\mathbf{x},u)}[L_d(D(E(\mathbf{x},u,\mathbf{A}))^\top D(E(\mathbf{x}',\mathbf{z}_j)), \mathbf{A}_{ij})]$$

$$= \underset{D}{\mathrm{argmin}}\, \mathbb{E}_{(\mathbf{e},j),(\mathbf{e}',j)\sim p(\mathbf{e},u)}[L_d(D(\mathbf{e})^\top D(\mathbf{e}'), \mathbf{A}_{ij})]$$

$$= \underset{D}{\mathrm{argmin}}\, \mathbb{E}_{(\mathbf{e},j),(\mathbf{e}',j)\sim p(\mathbf{e},u)}[\mathbf{A}_{ij}\log(\sigma(D(\mathbf{e})^\top D(\mathbf{e}'))) + (1 - \mathbf{A}_{ij})\log(1 - \sigma(D(\mathbf{e})^\top D(\mathbf{e}')))]$$

$$= \underset{D}{\mathrm{argmin}}\, \mathbb{E}_{\mathbf{e},\mathbf{e}'\sim p(\mathbf{e})}\mathbb{E}_{i\sim p(u|\mathbf{e}),j\sim p(u|\mathbf{e}')}[\mathbf{A}_{ij}\log(\sigma(D(\mathbf{e})^\top D(\mathbf{e}'))) + (1 - \mathbf{A}_{ij})\log(1 - \sigma(D(\mathbf{e})^\top D(\mathbf{e}')))]$$

$$= \underset{D}{\mathrm{argmin}}\, \mathbb{E}_{\mathbf{e},\mathbf{e}'\sim p(\mathbf{e})}[\alpha(\mathbf{e},\mathbf{e}')\log(\sigma(D(\mathbf{e})^\top D(\mathbf{e}'))) + (1 - \alpha(\mathbf{e},\mathbf{e}'))\log(1 - \sigma(D(\mathbf{e})^\top D(\mathbf{e}')))]$$

where $\alpha(\mathbf{e},\mathbf{e}') = \mathbb{E}_{i\sim p(u|\mathbf{e}),j\sim p(u|\mathbf{e}')}[\mathbf{A}_{ij}]$. Notice that the global minimum is achieved if for any $(\mathbf{e},\mathbf{e}')$, $\sigma(D(\mathbf{e})^\top D(\mathbf{e}'))$ minimize the negative binary cross entropy $\alpha(\mathbf{e},\mathbf{e}')\log(\sigma(D(\mathbf{e})^\top D(\mathbf{e}'))) + (1 - \alpha(\mathbf{e},\mathbf{e}'))\log(1 - \sigma(D(\mathbf{e})^\top D(\mathbf{e}')))$, or equivalently, $\sigma(D(\mathbf{e})^\top D(\mathbf{e}')) = \alpha(\mathbf{e},\mathbf{e}')$. □

Here we will use the chain of 3 nodes as a simple explanatory example: domain 2 is connected to both domain 1 and domain 3. Suppose data are mapped to either of the 2 encodings, $\mathbf{e} = 0$ and $\mathbf{e} = 1$. The probability of domain index $u$ conditioned on encoding $\mathbf{e}$ (i.e., $p(u|\mathbf{e})$) is given in Table 4. The discriminator will output the reconstruction result based on what encoding it obtains. For example, suppose we have two data points, $x_1$ and $x_2$, with their encodings $\mathbf{e}_1 = 0$ and $\mathbf{e}_2 = 1$. Their corresponding domain indices $u_1$ and $u_2$ given $\mathbf{e}_1$ and $\mathbf{e}_2$ follow the distributions $p(u|\mathbf{e} = 0)$ and $p(u|\mathbf{e} = 1)$, respectively. Note that here $\mathbf{e}_1$ and $\mathbf{e}_2$ correspond to $\mathbf{e}$ and $\mathbf{e}'$ in Lemma 4.1, while $u_1$ and $u_2$ correspond to $i$ and $j$ in Lemma 4.1. Based on such a realization, we can obtain $\mathbb{E}_{u_1\sim p(u|\mathbf{e}=0), u_2\sim p(u|\mathbf{e}=1)}[\mathbf{A}_{u_1 u_2}] = 0.38$ (assuming no self-loop in the domain adjacency graph). To minimize the loss, the optimal discriminator will then try to output $\sigma(D(\mathbf{e}_1)^\top D(\mathbf{e}_2)) = 0.38$.

**Theorem 4.1** (**Global Optimum for GRDA**). *Given an adjacency matrix $\mathbf{A}$, the total loss $L_d(D, E)$ has a tight upper bound:*

$$L_d(D, E) \leq H(\mathbb{E}_{\mathbf{e},\mathbf{e}'}\alpha(\mathbf{e},\mathbf{e}')) = H(\mathbb{E}_{i,j}[\mathbf{A}_{ij}]),$$

*where $H$ denotes the entropy function, $H(p) = -p\log(p) - (1 - p)\log(1 - p)$, and $\alpha(\mathbf{e},\mathbf{e}') = \mathbb{E}_{i\sim p(u|\mathbf{e}),j\sim p(u|\mathbf{e}')}[\mathbf{A}_{ij}]$. Furthermore, the equality, i.e., the optimum, is achieved when*

$$\alpha(\mathbf{e},\mathbf{e}') = \mathbb{E}_{i,j}[\mathbf{A}_{ij}], \text{ for any } e, e',$$

*or equivalently, $\mathbb{E}_{i,j}[\mathbf{A}_{ij}|\mathbf{e},\mathbf{e}'] = \mathbb{E}_{i,j}[\mathbf{A}_{ij}]$, where $(e, e')$ is from the encoding space.*

*Proof.* Denoting the discriminator output of the encoding $\mathbf{e}$ as $\mathbf{d_e}$, i.e, $\mathbf{d_e} = D(\mathbf{e})$, we have

$$L_d(D, E) = \mathbb{E}_{(\mathbf{e},i),(\mathbf{e}',j) \sim p_E(\mathbf{e},u)}[-\mathbf{A}_{ij} \log(\sigma(\mathbf{d_e}^\top \mathbf{d_{e'}})) - (1 - \mathbf{A}_{ij}) \log(1 - \sigma(\mathbf{d_e}^\top \mathbf{d_{e'}}))]$$

$$= \mathbb{E}_{(\mathbf{e},i),(\mathbf{e}',j)}[-\mathbf{A}_{ij} \log(\alpha(\mathbf{e}, \mathbf{e}')) - (1 - \mathbf{A}_{ij}) \log(1 - \alpha(e, e'))] \tag{6}$$

$$= \mathbb{E}_{\mathbf{e},\mathbf{e}'}[-\alpha(\mathbf{e}, \mathbf{e}') \log(\alpha(\mathbf{e}, \mathbf{e}')) - (1 - \alpha(\mathbf{e}, \mathbf{e}')) \log(1 - \alpha(\mathbf{e}, \mathbf{e}'))] \tag{7}$$

$$= \mathbb{E}_{\mathbf{e},\mathbf{e}'} H(\alpha(\mathbf{e}, \mathbf{e}')) \tag{8}$$

$$\leq H(\mathbb{E}_{\mathbf{e},\mathbf{e}'} \alpha(\mathbf{e}, \mathbf{e}')) = H(\mathbb{E}_{i,j}[\mathbf{A}_{ij}]), \tag{9}$$

Eqn. 6 and Eqn. 7 are due to the optimal discriminator, i.e., Lemma 4.1. The inequality, i.e., Eqn. 9, is due to the concavity of $H$ and Jensen's inequality. The optimum achieved when $\alpha(\mathbf{e}, \mathbf{e}') = \mathbb{E}_{i,j}[\mathbf{A}_{ij}]$ for any $\mathbf{e}, \mathbf{e}'$, i.e. $\mathbb{E}_{i,j}[\mathbf{A}_{ij}|\mathbf{e}, \mathbf{e}'] = \mathbb{E}_{i,j}[\mathbf{A}_{ij}]$. □

**Corollary 4.1.** *For GRDA, the global optimum of total loss $L_d(D, E)$ is achieved if the encoding of all domains (indexed by $u$) are perfectly (uniformly) aligned, i.e., $\mathbf{e} \perp u$.*

*Proof.* From Theorem 4.1, we have:

$$L_d(D, E) = \mathbb{E}_{\mathbf{e},\mathbf{e}'} H(\alpha(\mathbf{e}, \mathbf{e}'))$$

$$= \mathbb{E}_{\mathbf{e},\mathbf{e}'} H(\mathbb{E}_{i \sim p(u|\mathbf{e}), j \sim p(u|\mathbf{e}')} \mathbf{A}_{ij})$$

$$= \mathbb{E}_{\mathbf{e},\mathbf{e}'} H(\mathbb{E}_{i \sim p(u), j \sim p(u)} \mathbf{A}_{ij}) \tag{10}$$

$$= H(\mathbb{E}_{i \sim p(u), j \sim p(u)}[\mathbf{A}_{ij}])$$

$$= H(\mathbb{E}_{i,j}[\mathbf{A}_{ij}]),$$

which achieves the global optimum. Note that Eqn. 10 is due to the perfect alignment condition, i.e., $p(u|\mathbf{e}) = p(u)$. □

**Lemma (Optimal Predictor).** *Given the encoder $E$, the prediction loss $V_p(F, E) \triangleq L_p(F(E(\mathbf{x}, u)), y) \geq H(y|E(\mathbf{x}, u, \mathbf{A}))$ where $H(\cdot)$ is the entropy. The optimal predictor $F^*$ that minimizes the prediction loss is*

$$F^*(E(\mathbf{x}, u, \mathbf{A})) = P_y(\cdot|E(\mathbf{x}, u, \mathbf{A})).$$

Assuming the predictor $F$ and the discriminator $D$ are trained to achieve their optimal losses, by Lemma , the three-player game, $\min_{E,F} \max_D L_f(E, F) - L_d(D, E)$, can be rewritten as following training procedure of the encoder $E$,

$$\min_E C(E) \triangleq H(y|E(\mathbf{x}, u, \mathbf{A})) - \lambda_d C_d(E), \tag{11}$$

Table 4: Values of $p(u|\mathbf{e})$ for different $\mathbf{e}$ and $u$.

| $u$ | $\mathbf{e} = 0$ | $\mathbf{e} = 1$ |
|---|---|---|
| 1 | 0.1 | 0.7 |
| 2 | 0.3 | 0.2 |
| 3 | 0.6 | 0.1 |

where $C_d(E) \triangleq \min_D L_d(E, D) = L_d(E, D_E^*)$.

**Theorem 4.2.** *Assuming $u \perp\!\!\!\perp y$, if the encoder $E$, the predictor $F$ and the discriminator $D$ have enough capacity and are trained to reach optimum, any global optimal encoder $E^*$ has the following properties:*

$$H(y|E^*(\mathbf{x}, u, \mathbf{A})) = H(y|\mathbf{x}, u, \mathbf{A}), \tag{12}$$

$$C_d(E^*) = \max_{E'} C_d(E'), \tag{13}$$

*where $H(\cdot|\cdot)$ denotes the conditional entropy.*

*Proof.* Since $E(\mathbf{x}, u, \mathbf{A})$ is a function of $\mathbf{x}$, $u$, and $\mathbf{A}$, by the data processing inequality, we have $H(y|E(\mathbf{x}, u, \mathbf{A})) \geq H(y|\mathbf{x}, u, \mathbf{A})$. Hence, $C(E) = H(y|E(\mathbf{x}, u, \mathbf{A})) - \lambda_d C_d(E) \geq H(y|\mathbf{x}, u, \mathbf{A}) - \lambda_d \max_{E'} C_d(E')$. The equality holds if and only if $H(y|\mathbf{x}, u) = H(y|E(\mathbf{x}, u, \mathbf{A}))$ and $C_d(E) = \max_{E'} C_d(E')$. Therefore, we only need to prove that the optimal value of $C(E)$ is equal to $H(y|\mathbf{x}, u, \mathbf{A}) - \lambda_d \max_{E'} C_d(E')$ in order to prove that any global encoder $E^*$ satisfies both Eqn. 12 and Eqn. 13.

We show that $C(E)$ can achieve $H(y|\mathbf{x}, u, \mathbf{A}) - \lambda_d \max_{E'} C_d(E')$ by considering the following encoder $E_0(\mathbf{x}, u, \mathbf{A}) = P_y(\cdot|\mathbf{x}, u, \mathbf{A})$. It can be examined that $H(y|E_0(\mathbf{x}, u, \mathbf{A})) = H(y|\mathbf{x}, u, \mathbf{A})$ and $E_0(\mathbf{x}, u, \mathbf{A}) \perp\!\!\!\perp u$ which leads to $C_d(E_0) = \max_{E'} C_d(E')$ using Corollary 4.1, completing the proof. $\qquad \square$

Note that $u \perp\!\!\!\perp y$ is a weak assumption because it can be true even if both $y \not\perp\!\!\!\perp u|\mathbf{x}, \mathbf{A}$ and $y \not\perp\!\!\!\perp \mathbf{x}|u, \mathbf{A}$ are true.

**Corollary 4.2** (**Cliques**). *In a clique, the GRDA optimum is achieved if and only if the embedding distributions of all the domains are the same, i.e. $p_1(\mathbf{e}) = \cdots = p_N(\mathbf{e}) = p(\mathbf{e}), \forall \mathbf{e}$.*

*Proof.* We start from Theorem 4.1. For any embedding pair $(\mathbf{e}, \mathbf{e}')$, we have $\frac{N(N-1)}{N^2} = \mathbb{E}[\mathbf{A}_{ij}] = \mathbb{E}[\mathbf{A}_{ij}|\mathbf{e}, \mathbf{e}'] = \frac{1}{N^2} \sum_{i<j} (\beta_i(\mathbf{e})\beta_j(\mathbf{e}') + \beta_i(\mathbf{e}')\beta_j(\mathbf{e}))$. First, consider the case of $\mathbf{e}' = \mathbf{e}$, which leads to $N(N-1) = \sum_{i<j} 2\beta_i(\mathbf{e})\beta_j(\mathbf{e})$. Since $\sum_i \beta_i(\mathbf{e}) = N$, we have $(\sum_i \beta_i(\mathbf{e}))^2 = N^2$. By the subtraction between two equations, we have $\sum_i \beta_i(\mathbf{e})^2 = (\sum_i \beta_i(\mathbf{e}))^2 - \sum_{i<j} 2\beta_i(\mathbf{e})\beta_j(\mathbf{e}) = N^2 - N(N-1) = N$. We can further have that $\sum_{i<j} (\beta_i(\mathbf{e}) - \beta_j(\mathbf{e}))^2 = (N-1)(\sum_i \beta_i(\mathbf{e})^2) - \sum_{i<j} 2\beta_i(\mathbf{e})\beta_j(\mathbf{e}) = (N-1)N - N(N-1) = 0$, leading to $\beta_i(\mathbf{e}) = \beta_j(\mathbf{e}), \forall i \neq j$. Considering $\sum_{i=1}^N \beta_i(\mathbf{e}) = N$, we have $\beta_i(\mathbf{e}) = 1, \forall i$. It is easily to see that the solution also satisfies the case of $\mathbf{e} \neq \mathbf{e}'$. $\qquad \square$

**Corollary 4.3** (**Star Graphs**). *In a star graph, the GRDA optimum is achieved if and only if the embedding distribution of the center domain is the average of all peripheral domains, i.e. $p_1(\mathbf{e}) = \frac{1}{N-1} \sum_{i=2}^N p_i(\mathbf{e}), \forall \mathbf{e}$.*

*Proof.* Similarly, we start from Theorem 4.1. For any embedding pair $(\mathbf{e}, \mathbf{e}')$, we have $\frac{2(N-1)}{N^2} = \mathbb{E}[\mathbf{A}_{ij}] = \mathbb{E}[\mathbf{A}_{ij}|\mathbf{e}, \mathbf{e}'] = \frac{1}{N^2} \sum_{i=2}^N (\beta_1(\mathbf{e})\beta_i(\mathbf{e}') + \beta_1(\mathbf{e}')\beta_i(\mathbf{e}))$. Leveraging the fact that $\sum_{i=1}^N \beta_i(\mathbf{e}) = N$, we have $2(N-1) = \beta_1(\mathbf{e})(N - \beta_1(\mathbf{e}')) + \beta_1(\mathbf{e}')(N - \beta_1(\mathbf{e}))$. Let's first consider the case of $\mathbf{e}' = \mathbf{e}$, we have $N-1 = \beta_1(\mathbf{e})(N - \beta_1(\mathbf{e}))$ which means that, for any $\mathbf{e}$, $\beta_1(\mathbf{e})$ is either $1$ or $N-1$. Now consider the case of $\mathbf{e}' \neq \mathbf{e}$ with the constraint that $\beta_1(\mathbf{e}), \beta_1(\mathbf{e}') \in \{1, N-1\}$; the solutions are $\beta_1(\mathbf{e}) = \beta_1(\mathbf{e}') = 1$ or $\beta_1(\mathbf{e}) = \beta_1(\mathbf{e}') = N-1$, for any pair $(\mathbf{e}, \mathbf{e}')$. Clearly it is not possible that $\alpha(\mathbf{e}) = N-1, \forall \mathbf{e}$, since it violates the equality $\mathbb{E}_{\mathbf{e}}[1|u=1] = \mathbb{E}_{\mathbf{e}}[1]$. Therefore the only solution is $\alpha(\mathbf{e}) = 1, \forall \mathbf{e}$, which implies that $p_1(\mathbf{e}) = \frac{1}{N-1} \sum_{i=2}^N p_i(\mathbf{e}), \forall \mathbf{e}$. $\qquad \square$

**Corollary 4.4** (**Chain Graphs**). *Let $p_i(\mathbf{e}) = p(\mathbf{e}|u=i)$ and the average encoding distribution $p(\mathbf{e}) = N^{-1} \sum_{i=1}^N p(\mathbf{e}|u=i)$. In a chain graph, the GRDA optimum is achieved if and only if $\forall \mathbf{e}, \mathbf{e}'$*

$$\sum_{i=1}^{N-1} \frac{p_i(\mathbf{e})p_{i+1}(\mathbf{e}') + p_i(\mathbf{e}')p_{i+1}(\mathbf{e})}{p(\mathbf{e})p(\mathbf{e}')} = 2(N-1).$$

*Proof.* By Theorem 4.1, $\forall \mathbf{e}, \mathbf{e}'$

$$2(N-1) = N^2 \mathbb{E}[\mathbf{A}_{ij}|\mathbf{e}, \mathbf{e}'] = N^2 \sum_{i=1}^N \sum_{j=1}^N \mathbf{A}_{ij} p(u=i|\mathbf{e})p(u=j|\mathbf{e}')$$

$$= N^2 \sum_{i=1}^N \sum_{j=1}^N \mathbf{A}_{ij} \frac{p_i(\mathbf{e})p(u=i)p_j(\mathbf{e}')p(u=j)}{p(\mathbf{e})p(\mathbf{e}')} = \sum_{i=1}^{N-1} \frac{p_i(\mathbf{e})p_{i+1}(\mathbf{e}') + p_i(\mathbf{e}')p_{i+1}(\mathbf{e})}{p(\mathbf{e})p(\mathbf{e}')},$$

which completes the proof. $\qquad \square$

**Proposition 4.1** (**Chain of Three Nodes**). *In this length three chain graph, the GRDA optimum is achieved if and only if the embedding distribution of the middle domain is the average of the embedding distributions of the domains on the two sides, i.e. $p_2(\mathbf{e}) = \frac{1}{2}(p_1(\mathbf{e}) + p_3(\mathbf{e})), \forall \mathbf{e}$.*

*Proof.* Since a chain graph of three nodes is also a star graph, we have that $p_2(\mathbf{e}) = \frac{1}{2}(p_1(\mathbf{e}) + p_3(\mathbf{e})), \forall \mathbf{e}$. $\qquad \square$

Table 5: Accuracy (%) on CompCars with Adagraph(Mancini et al., 2019) as backbone encoders. In this experiment, we used the network proposed by Adagraph and trained it with adversarial domain adaptation methods. We mark the best result with **bold face** and the second best results with underline.

| Method | Ada Only | Ada + DANN | Ada + ADDA | Ada + CDANN | Ada + MDD | Ada + GRDA (Ours) |
|---|---|---|---|---|---|---|
| *CompCars* | $55.05 \pm 0.87$ | $55.87 \pm 1.22$ | $55.12 \pm 1.28$ | $55.82 \pm 1.16$ | $\mathbf{56.63} \pm \underline{0.64}$ | $\underline{56.56} \pm \mathbf{0.38}$ |

## B  ADAGRAPH + ADVERSARIAL LEARNING METHODS

Tabel 5 shows the results when using Adagraph(Mancini et al., 2019) as the backbone encoders for different adversarial methods on the CompCars image classification task. From the table, we could see that adversarial methods consistently improve the adaptation performance, which demonstrates the orthogonality of Adagraph and adversarial methods. In this experiment, our method GRDA achieves high accuracy and stability (the lowest standard deviation among all adversarial methods).

## C  SEMI-SUPERVISED METHODS

At first blush, our problem setting may look similar to semi-supervised learning (SSL) on graphs, which can be handled by graph neural networks (GNN), e.g., GCN (Kipf & Welling, 2016a) and GAT (Velivcković et al., 2017). However, unsupervised domain adaptation (UDA) problem and SSL are very different from various perspectives.

First, in semi-supervised learning (SSL) problems, the focus is typically on the whole predictor and it often dictates smoothness of the predictor w.r.t. the graph. In contrast, graph-relational domain alignment (GRDA) focuses on the embeddings (latent representations), and allows a much more flexible predictor. For example, the predictor in GRDA does not have to be smooth w.r.t. the graph.

Second, these two problems have very different assumptions. Specifically, UDA often assumes domain shift and tries to align data from different domains before learning a predictor. In contrast, SSL does not assume domain shift and directly uses unlabeled data to improve the decision boundary. Therefore, when domain shift exists, formulating the problem as UDA often leads to better accuracy.

Both our theoretical analysis and empirical results on DG-15 as well as DG-60 show that typical GNN-SSL methods do not work. Specifically, in this section we discuss 2 variants of GNN-SSL methods adapted for our setting:

- **Variant 1 (GNN-SSL-V1):** We treat each sample as a node and connect two samples if their domains are the same or immediately adjacent. We then perform node classification on samples.

- **Variant 2 (GNN-SSL-V2):** We treat each domain as a node and treat each sample as a node as well. Domain nodes connect to adjacent domain nodes while sample nodes only connect to domain nodes that they belong to. We then perform node classification on samples.

For Variant 1, if we use Graph Convolutional Network (GCN) as our GNN, we can prove the following theorem:

**Theorem C.1 (Negative Results on Variant 1).** *For Variant 1, every node (data point) from the same domain will have the same node embedding after applying GCN.*

*Proof.* For GCN, the $l$-th neural network layer will perform the operation $h_i^{(l+1)} = \sigma(b^{(l)} + \sum_{j \in \mathcal{N}(i)} \frac{1}{c_{ji}} h_j^{(l)} W^{(l)})$, where $\mathcal{N}(i)$ contains the neighbours of node $i$, $c_{ij} = \sqrt{|\mathcal{N}(j)|}\sqrt{|\mathcal{N}(i)|}$, and $\sigma$ is an activation function. $h_i^{(l)}$ is the output of the $(l-1)$-th graph convolution layer ($h_i^{(0)}$ is the node feature). $W^{(l)}$ and $b^{(l)}$ denote weight and bias of the $l$-th layer, respectively (Kipf & Welling, 2016a).

Table 6: Accuracy (%) on *DG-15* and *DG-60*.

| Method | Source-Only | **GNN-SSL-V1** | **GNN-SSL-V2** | DANN | ADDA | CDANN | MDD | **GRDA (Ours)** |
|---|---|---|---|---|---|---|---|---|
| *DG-15* | 39.77 | 50.00 | 60.15 | 72.11 | 69.00 | 72.44 | 51.33 | **84.44** |
| *DG-60* | 38.39 | 50.00 | 35.07 | 61.98 | 32.17 | 61.70 | 66.24 | **96.76** |

Table 7: Accuracy (%) on *DG-15* and *DG-60*.

| Method | Source-Only | **SENTRY** | DANN | ADDA | CDANN | MDD | GRDA (Ours) |
|---|---|---|---|---|---|---|---|
| *DG-15* | 39.77 | 43.67 | 72.11 | 69.00 | 72.44 | 51.33 | **84.44** |
| *DG-60* | 38.39 | 47.80 | 61.98 | 32.17 | 61.70 | 66.24 | **96.76** |

Table 8: Accuracy (%) on TPT-48.

| Method | Source-Only | **SENTRY** | DANN | ADDA | CDANN | MDD | GRDA (Ours) |
|---|---|---|---|---|---|---|---|
| *E (24)→W (24)* | 1.926 | 2.73 | 2.679 | 2.254 | 2.108 | 1.335 | **1.297** |
| *N (24)→S (24)* | 2.426 | 9.44 | 2.600 | 2.709 | 2.621 | 2.054 | **1.165** |

Table 9: Accuracy (%) on CompCars (4-Way Classification).

| Method | Source-Only | **SENTRY** | DANN | ADDA | CDANN | MDD | GRDA (Ours) |
|---|---|---|---|---|---|---|---|
| *CompCars* | 46.5 | 31.39 | 50.2 | 46.1 | 48.2 | 49.0 | **51.0** |

Based on the definition of this method, if data points $k$ and $m$ belong to the same domain, they will have $\mathcal{N}(k) = \mathcal{N}(m) = \mathcal{N}$ (each node has a self loop). Therefore we have

$$h_k^{(l+1)} - h_m^{(l+1)} = \sigma(\sum_{j \in \mathcal{N}} (\frac{1}{c_{jk}} h_j^{(l)} - \frac{1}{c_{jm}} h_j^{(l)}) W^{(l)})$$

$$= \sigma(\sum_{j \in \mathcal{N}} (\frac{1}{\sqrt{|\mathcal{N}(j)|}\sqrt{|\mathcal{N}(k)|}} h_j^{(l)} - \frac{1}{\sqrt{|\mathcal{N}(j)|}\sqrt{|\mathcal{N}(m)|}} h_j^{(l)}) W^{(l)}) = 0,$$

completing the proof. □

Theorem C.1 indicates that if we use GCN, the data points in the same domain will obtain same prediction result (because their node embeddings are the same). This explains the experiment results in Table 6. Specifically, for binary classification tasks on DG-15 and DG-60, the number of positive and negative samples are identical in each domain. The polarized prediction (samples in the same domain are predicted as either all positive or all negative) leads to an accuracy of exactly 50%. Although we use GCN as our graph neural network during derivation, we can easily generalize our conclusion to other GNNs as long as they follow similar neighbour aggregation strategies.

For Variant 2 (GCN-SSL-V2), we conduct experiments on both DG-15 and DG-60 as well. As shown in the Table 6, GCN-SSL-V2 methods perform worse than most domain adaptation baselines on DG-15, and do not even outperform Souce-Only on DG-60.

## D   ENTROPY-BASED DOMAIN ADAPTATION METHODS

For completeness, we include SENTRY(Prabhu et al., 2021), a recent entropy-based method as a baseline. Essentially, entropy-based methods perform self-training by minimizing the conditional entropy of model predictions, making the predictions more confident on the target domain. Naturally such a training strategy can only work on single-source single-target adaptation. To adapt SENTRY to our tasks, we combine all the data in multiple source domains into a single source domain, and similarly for multiple target domains. On DG-15 and DG-60 (Table 7), SENTRY only outperforms Source-Only; on TPT-48 (Table 8) and CompCars (Table 9), SENTRY even underperforms Source-Only, possibly due to the mixture of source data and target data.

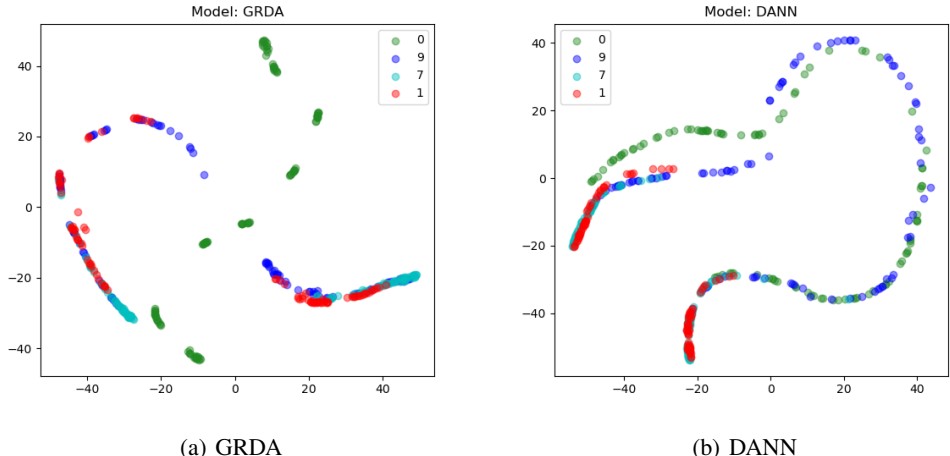

(a) GRDA                                    (b) DANN

Figure 8: Visualization of the encodings of data produced on DG-15.

Table 10: Accuracy of GRDA and DANN on DG-15.

| # Source Domains | 1 | 2 | 4 | 6 | 8 |
|---|---|---|---|---|---|
| GRDA | 64.7 | 74.1 | 81.5 | 84.4 | 100.0 |
| DANN | 42.1 | 44.6 | 33.5 | 72.11 | 88.7 |

## E    VISUALIZATION OF ENCODINGS

We also visualize the encodings of data produced on DG-15 here. All the encodings are reduced to 2 dimensions by TSNE.

Figure 9(a) shows the domain graph for DG-15. To showcase GRDA's ability to align encodings according to the domain graph, Figure 8(a) plots the encodings of domains 0, 1, 7, and 9 produced by GRDA. We can see the encodings from adjacent domains (i.e., domains 9, 7, and 1) align with each other, while the encodings from distant domains (i.e., domains 0 and 9) do not align with each other. This demonstrates that our model successfully relaxes uniform alignment according to the domain graph.

Figure 8(b) shows the encodings of DANN. We can see that although domain 0 and 9 are distant, they still align with each other, showing that classical domain adaptation methods lack the flexibility of alignment compared with GRDA.

## F    DATASET VISUALIZATION

Figure 9(a) and 9(b) show the angle $\omega_i = \arcsin(\frac{b_i}{a_i})$ of each domain for *DG-15* and *DG-60* (the angle of each ground-truth boundary is $\omega_i$). Figure 10 visualizes a part of the *TPT-48* dataset that is used for training and testing.

## G    THE EFFECT OF SOURCE DOMAIN NUMBERS

Table 10 provides results on *DG-15* for different numbers of source domains. It shows that the accuracy increases with more source domains, and that GRDA still outperforms the baselines in all cases.

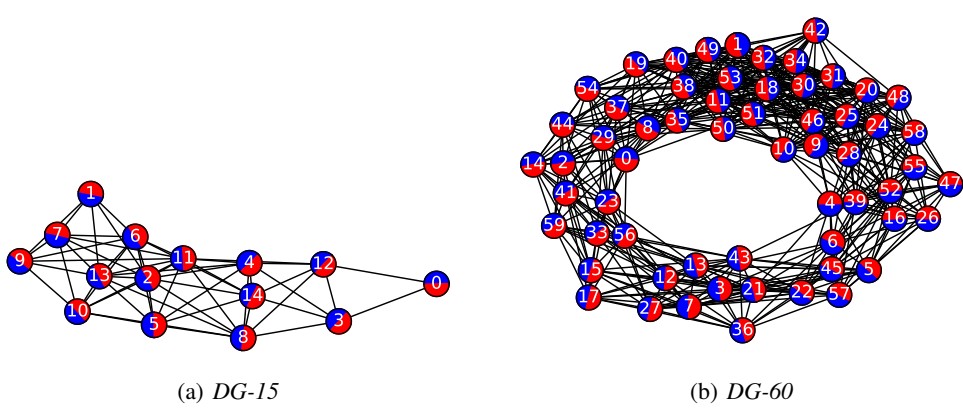

(a) *DG-15*                    (b) *DG-60*

Figure 9: Visualization of the *DG-15* (left) and *DG-60* (right) datasets. We use 'red' and 'blue' to roughly indicate positive and negative data points inside a domain. The boundaries between 'red' half circles and 'blue' half circles show the direction of ground-truth decision boundaries in the datasets.

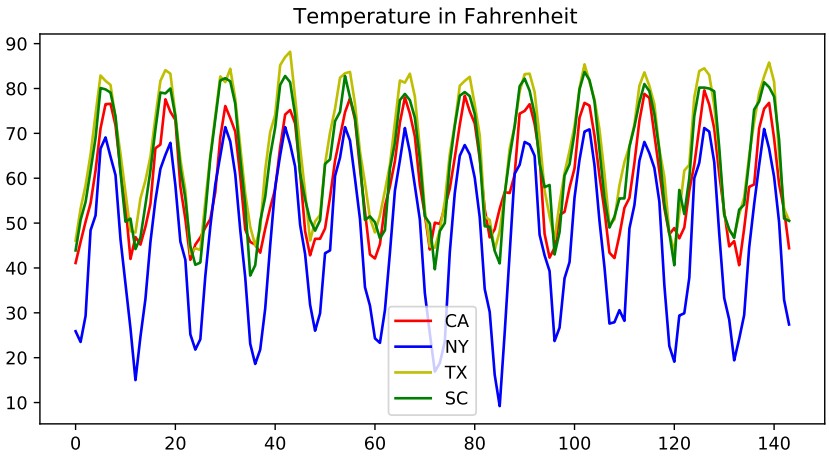

Figure 10: Visualization for data of four of the states in *TPT-48*. Here we show the states' monthly average temperature in Fahrenheit.

## H  IMPLEMENTATION DETAILS

### H.1  MODEL ARCHITECTURES

To ensure fair comparison, all algorithms use the same network architecture for the encoder and the predictor. The encoder of each model consists of 3 components.

- A **graph encoder** embeds the domain graph $A$ and the domain index $u_l$ to the domain embeddings $e_l$.
- A **raw data encoder** embeds the data $x_l$ into data embeddings $h_l$.
- A **joint encoder** then takes as input both $e_l$ and $h_l$ and produces the final embeddings.

For *DG-15*, *DG-60* and *TPT-48*, the raw data encoder contains 3 fully connected (FC) layers, and the predictor contains 3 FC layers, both with ReLU as activation functions. For *CompCars*, we use AlexNet (Krizhevsky et al., 2012) as the raw data encoder. All joint encoders contain 2 FC layers.

The discriminators of different algorithms all have 6 FC layers, with slight differences on the output dimension. GRDA's discriminator produces a $k$-dimensional node (domain) embedding.

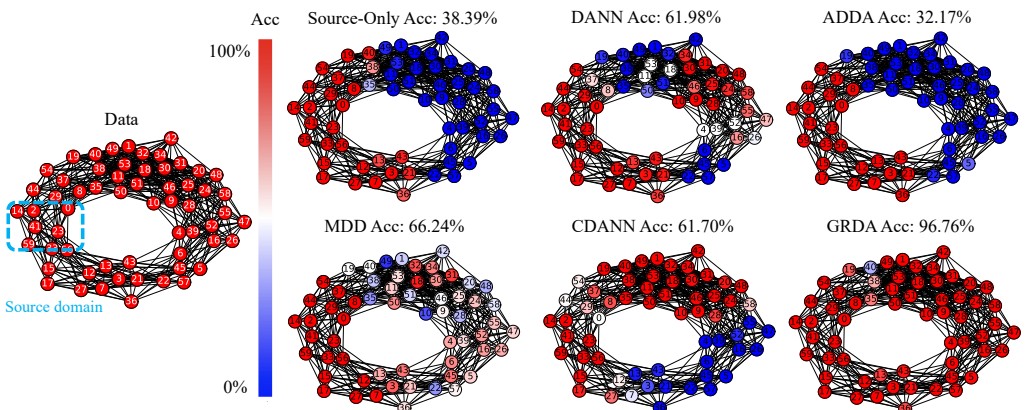

Figure 11: Detailed results on *DG-60* with 60 domains. On the left is the domain graph for *DG-60*. We use the 6 domains in the dashed box as source domains. On the right is the accuracy of various DA methods for each domain, where the spectrum from 'red' to 'blue' indicates accuracy from 100% to 0% (best viewed in color).

## H.2 OTHER HYPERPARAMETERS

For experiments on all 4 datasets, we choose $k = 2$. We use a mixture policy for sampling nodes (domains) to train GRDA's discriminator. One method is to randomly sample several nodes, and another is to pick the nodes from randomly chosen connected sub-graphs. We pick one of the policies randomly in each iteration and calculate the loss of each forward pass. The models are trained using the Adam (Kingma & Ba, 2015) optimizer with learning rates ranging from $1 \times 10^{-5}$ to $1 \times 10^{-4}$, and $\lambda_d$ ranging from 0.1 to 1.For each adaptation task, the input data is normalized by its mean and variance. We run all our experiments on a Tesla V100 GPU using AWS SageMaker (Liberty et al., 2020).

## H.3 TRAINING PROCESS

We perform the standard gradient-based alternating optimization for minimax games (e.g., DANNs and GANs); we iteratively perform the following 2 steps: **(a)** optimizing discriminator $D$ with the encoder $E$ and predictor $F$ fixed, and **(b)** optimizing encoder $E$ and predictor $F$ with the discriminator $D$ fixed. Specifically:

For **(a)**, we first use the encoder mentioned above to produce the encoding and then use the loss function in Equation (3) of the main paper to train the discriminator $D$. This loss function quantifies whether the node embedding reproduced by D preserves domain connection information in $A$.

For **(b)**, we fix the discriminator $D$ and minimize the predictor loss plus the negation of the discriminator loss (i.e., $L_f(E, F) - \lambda_d L_d(D, E)$) to train the encoder $E$ and the predictor $F$. This loss function enables the encoder to preserve useful features for prediction while removing domain-related information in the encoding to align different domains in the encoding space.

We perform these 2 steps iteratively until convergence.

## I DETAILED RESULTS FOR EACH DOMAIN ON *DG-60* AND *CompCars*

Figure 11 shows the detailed results on *DG-60*. It shows that GRDA significantly outperforms all baselines. We also include the detailed results on *CompCars* (Figure 12), where GRDA also outperforms all the other methods.

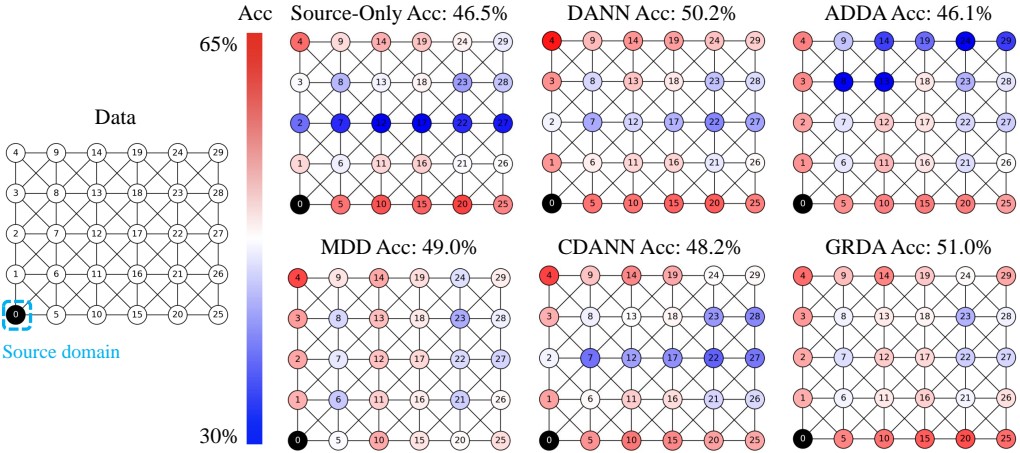

Figure 12: Detailed results on *CompCars* with 30 domains. On the left is the domain graph for *CompCars*. We use the domain in the dashed box as the source domain. On the right is the accuracy of various DA methods for each domain, where the spectrum from 'red' to 'blue' indicates accuracy from 65% to 30% (best viewed in color).

