# OpenReview forum: "Graph-Relational Domain Adaptation"
_ICLR.cc/2022/Conference — ICLR 2022 Poster_

### Official Review · Reviewer_uDYW · 2021-10-30

**Correctness:** 3
**Technical Novelty And Significance:** 3
**Empirical Novelty And Significance:** 3
**Recommendation:** 6
**Confidence:** 3

**Main Review:**


Strengths:
1.	This paper propose novel method multi-target domain adaptation. Authors introduce the adversarial learning method for this issue, which looks interesting.
2.	The paper gives clear description about this problem, and the proposed method, which is easy to follow.
3.	Using varying kinds of datasets to test the provided method, which is solid.
4.	I like the shown figure in this paper, which is great for me to depict how the proposed method works.

Weaknesses:
1.	Do this proposed method for traditional domain adaptation with two domain. What I mean is to learn the graph for data instead of domain.
2.	The reported results looks has little advantage compared to baselines. Do authors pay more energy and parameters ?


--------------------After rebuttal----------------------

Thanks for authors' response. Authors have addresses my concerns excepting for the further experiment for two domain, which does not influence my score. After reading other reviewers' comments (e.g., on one mention the less novelty), it think the proposed method still keep advantages compared existing methods.  I would like  keep my score.

**Summary Of The Paper:**

This paper explore multi-target domain adaptation by graph.  For this setting, It is not suitable to treat equally every domain, and align them all perfectly.  To remedy this problem, In this paper authors propose the graph-based method to learn the adjacency between domains.  In this paper, authors propose adversarial learning method, which refers the encoder and the discriminator. What is more,  authors show the theoretical analysis for this problem. The reported results shows that the proposed method successfully generalizes uniform alignment.

**Summary Of The Review:**

It looks the proposed method is new, since it built an adversarial learning for this question.

---

> ### Author Response · Authors · 2021-11-16
> **Response for your questions on our method, results, etc.**
>
> Thanks for your valuable comments! We are glad that you find our method novel, our experiments solid, our paper well-written and our figure helpful. Below we address your questions one by one.
>
> **Q1: Do this proposed method for traditional domain adaptation with two domains. What I mean is to learn the graph for data instead of domain.**
>
> Thanks for your suggestion! So far GRDA assumes that a domain graph is available. Inferring the graph in the data-instance level is definitely interesting future work. With the inferred graph, our GRDA can potentially treat each data instance as a domain and directly perform adaptation.
>
> **Q2: The reported results have little advantage compared to baselines. Do authors pay more energy and parameters ?**
>
> We put much effort in parameter tuning for both our method and baselines. A grid search is applied to the learning rate and $\lambda_d$ (see Appendix F.2 for details). Our method outperforms baselines by large margins on toy examples; in real world datasets, the improvement margins are still significant, though smaller.
>
> Specifically, our results on Table 1, 2, and 3 show that our model outperforms SOTA baselines including DANN, CDANN, ADDA, and MDD. **(1)** In Table 1, we showed that GRDA outperforms the baselines by large margins (>10% in DG-15 and >30% in DG-60). **(2)** In Table 2, we showed that the performance improvement from Source-Only to GRDA is three times the improvement from Source-Only to the best baseline MDD in the N->S case. **(3)** Table 3 shows, in the CompCar dataset, GRDA achieves 0.8% - 4% accuracy improvements on the baselines. This improvement is already significant. For instance,  (Tran et al., 2019; Wang et al., 2020b) report a 2-4% accuracy improvement on the CompCar dataset. These experiments verify that the proposed graph discriminator is better than the original discriminator and its variants.
>
> We think there are 2 possible reasons for the smaller improvement margins in real-world datasets.
>
> **(1)** The graph in the toy example is the ground-truth graph. In some complicated, real-world applications, ``ground-truth’’ graphs are not available, and we can only estimate it. The estimation error simply propagates. However, we observe that graph-relational domain adaptation can yield practical improvements even with the estimated graph. We believe with more information, we can estimate the graph more accurately, thereby further improving the accuracy. For example, in TPT-48, we use geographic adjacency to construct the domain graph; other information such as flight frequency between states can be incorporated to construct a better domain graph and improve the performance. Besides, as written in the paper and the answer of Q1, incorporating graph estimation into the learning problem is indeed an important future direction.
>
> **(2)** The complexity of the prediction problem also affects the performance margin. For toy datasets, we have a clear boundary to separate positive and negative data. Therefore, if the model finds the correct way to align data from different domains, the performance will improve significantly. For CompCars, the problem is much more complex and the data is much noisier. In this case, it is not easy for the encoder to map the data into the optimal feature space where data can be perfectly separated. Therefore the improvement tends to be smaller. Nevertheless, we note that our GRDA still consistently improves upon all the baselines.
>
>
> Yuchen Zhang, Tianle Liu, Mingsheng Long, and Michael I Jordan. Bridging theory and algorithm for domain adaptation. arXiv preprint arXiv:1904.05801, 2019
>
> Sinan Wang, Xinyang Chen, Yunbo Wang, Mingsheng Long, and Jianmin Wang. Progressive adversarial networks for fine-grained domain adaptation. In Proceedings of the IEEE/CVF conference on computer vision and pattern recognition, pp. 9213–9222, 2020b.

---

> ### Author Response · Authors · 2021-11-30
> **Thank you**
>
> Thank you for your positive and constructive feedback as well as the further response! We are glad that you have a thorough understanding of our contribution, find it novel, and acknowledge our improvement upon SOTA methods. We would be immensely grateful if you could raise the confidence score to reflect the current assessment.
>
> Thank you!
>
> Bests,
>
> GRDA authors

---

### Official Review · Reviewer_rNQp · 2021-11-01

**Correctness:** 3
**Technical Novelty And Significance:** 3
**Empirical Novelty And Significance:** 2
**Recommendation:** 5
**Confidence:** 5

**Details Of Ethics Concerns:**

See Main Review.

**Main Review:**

This work includes some advantages:
1. The authors generalize the adversarial learning framework by replacing the domain discriminator with a graph discriminator.
2. Some theoretical analyses support their claims on such a replacement.
3. Experimental results validate the usefulness of the proposed method on domain adaption.

Some concerns are listed as follows.
1. Domain alignment has been intensively studied and graph-based domain adaptation also has been well discussed in the existing works.  There are lots of variants on the adversarial discriminator or entropy-based or distribution alignment theories. The authors fail to convince the reviewer the proposed graph discriminator is better than the original adversarial discriminator or its variants, although they present a series of theorems and corollaries in the support section. To me, adversarial discriminators are more general than the given graph discriminator.
2. Graph-based discriminator has appeared in the existing works. The authors should give the reviewer a full explanation of your contributions.
3. For experiments, there are many advanced SOTA works, while the authors mainly compared some conventional methods, which are not the SOTA domain adaption works.
4. If the proposed method can replace the original adversarial works, this paper could be rated a high score for acceptance. The authors should pay more attention to the advantages of the proposed methods, and how could you make your work outperform these popular works, which is also the goal of ICLR.

**Summary Of The Paper:**

This work optimizes the adversarial domain adaption model by generalizing the uniform domain alignment with a domain encoding graph. The proposed method is simple and easy to follow, and the overall idea is clearly described. Some experiments show the effectiveness of the proposed method on both synthetic and real-world datasets.

**Summary Of The Review:**

This paper proposes a new graph discriminator to replace the conventional adversarial discriminator in adversarial domain adaptation. Some theoretical and experimental analyses validate their claims, while some concerns should be clarified and improved.

---

> ### Author Response · Authors · 2021-11-16
> **Response for your questions on existing works, baselines, etc.**
>
> Thank you for your valuable reviews. We are glad that you find our theoretical analysis supportive and our method useful. Below we address your questions one by one.
>
> **Q1: Domain alignment has been intensively studied and graph-based domain adaptation also has been well discussed in the existing works. There are lots of variants on the adversarial discriminator or entropy-based or distribution alignment theories. The authors fail to convince the reviewer the proposed graph discriminator is better than the original adversarial discriminator or its variants, although they present a series of theorems and corollaries in the support section. To me, adversarial discriminators are more general than the given graph discriminator.**
>
> First, our results on Table 1, 2, and 3 have shown that our model outperforms the original adversarial discriminator or its variants including DANN, CDANN, ADDA, and MDD. **(1)** In Table 1, we showed that GRDA outperforms the baselines by large margins (>10% in DG-15 and >30% in DG-60). **(2)** In Table 2, we showed that the performance improvement from Source-Only to GRDA is three times the improvement from Source-Only to the best baseline MDD in the N->S case. **(3)** In Table 3, the GRDA improves the accuracy by about 4% in the CompCar dataset. These experiments verify that the proposed graph discriminator is better than the original discriminator and its variants. The other reviews agree that our experiments are convincing. Reviewer uDYW commented that we ``"varying kinds of datasets to test the provided method, which is solid"``. Reviewer n4Lk commented that ``"experiments are well thought out and highlight the parameter study which makes the results of the work reproducible"``.
>
> Second, our graph discriminator belongs to the broad category of adversarial discriminator.
> Typical adversarial discriminators directly classify domain indices, which might look general but achieved a low performance because they neglected the domain graph. Our method, however, takes full advantage of the domain graph and thus provides better performance. The other reviews agree that our method is valuable and novel. Reviewer n4Lk commented that ``"the idea of using a discriminator to reconstruct rather than classify is novel"``. Reviewer uDYW commented that we propose a ``"novel method...which looks interesting"``.
>
>
> **Q2: Graph-based discriminator has appeared in the existing works. The authors should give the reviewer a full explanation of your contributions.**
>
> We have given a detailed literature review and comparison in Section 2. Many works focused on adaptation between two domains where data points themselves are graphs, e.g.,  (Pilanci & Vural, 2019; Pilanci & Vural, 2020) as cited in our paper, but their domains do not live on a graph and they are not graph discriminators. Our model focuses on a totally different setting where each domain is a node in the graph. As far as we know, there have been no previous works that build a discriminator to reconstruct the graph. Reviewer n4Lk also agreed and commented that ``"the idea of using a discriminator to reconstruct rather than classify is novel"``.
>
> If you have any particular references in mind, we will be happy to cite them and explain the contributions and differences compared with our work.
>
> **Q3: For experiments, there are many advanced SOTA works, while the authors mainly compare some conventional methods, which are not the SOTA domain adaptation works.**
>
> One of our baseline **MDD** (Margin Disparity Discrepancy) comes from ICML 2019, which is a popular and commonly used SOTA method and is different from the traditional model **MMD** (Maximum Mean Discrepancy). We believe that all the methods chosen are representative, and we expect other methods have similar performance. If you have any particular references in mind, we will be very happy to include them as additional baselines and update the results during the discussion period.
>
> **Q4: If the proposed method can replace the original adversarial works, this paper could be rated a high score for acceptance. The authors should pay more attention to the advantages of the proposed methods ...**
>
> Thanks for your suggestion. Our method can replace the original adversarial works, because they are actually special cases of our method, that is, they are *equivalent* to our GRDA with a clique domain graph (Corollary 4.2). This means that without the domain graph, our model has the same capability as the original adversarial works, and that when a domain graph is available, with the flexibility of alignment, we can achieve better performance. We show empirical evidence in experiments on DG-15, DG-60, TPT-48 and CompCars. The other reviews agree that our method outperforms SOTA baselines. Reviewer n4Lk commented that ``"empirical results on both synthetic and real-world datasets demonstrate the superiority of this method"``.

---

> ### Author Response · Authors · 2021-11-26
> **Response for your questions on entropy-based baselines.**
>
> According to your suggestion, we have added a new state-of-the-art entropy-based method, SENTRY (Prabhu et al., 2021) from ICCV 2021 as a baseline.  The following table shows the results for SENTRY on two datasets, DG-15 and DG-60. We can see that while SENTRY is better than Source-Only and some other baselines such as ADDA (on DG-60), our proposed GRDA can still outperform SENTRY by a large margin.
>
> Note that entropy-based methods like SENTRY only work for classification tasks; therefore it is not applicable to TPT-48, our real-world regression dataset. We are currently running experiments on CompCar, our real-world classification dataset, and will post the results once they are available.
>
>
> | Method | Source-Only | DANN | ADDA | CDANN | MDD | SENTRY | GRDA (Ours) |
> | :----: | :----: |:----: |:----: |:----: |:----: | :----: | :----: |
> | DG-15 | 39.77 | 72.11 | 69.00 | 72.44 | 51.33 | 43.67 | 84.44 |
> | DG-60 | 38.39 | 61.98 | 32.17 | 61.70 | 66.24 | 47.80 | 96.76 |
> | | | | | | |
>
>
> Viraj Prabhu, Shivam Khare, Deeksha Kartik, and Judy Hoffman.  Sentry: Selective entropy optimization via committee consistency for unsupervised domain adaptation. In Proceedings of the IEEE/CVF International Conference on Computer Vision, pp. 8558–8567, 2021

---

### Official Review · Reviewer_n4Lk · 2021-11-02

**Correctness:** 3
**Technical Novelty And Significance:** 2
**Empirical Novelty And Significance:** Not applicable
**Recommendation:** 6
**Confidence:** 4

**Main Review:**

Some specific comments are as follows:

Pros:
- Paper presents interesting ideas on how to utilize the original topological structures among different domains to benefit the domain adaptation results. The idea of using a discriminator to reconstruct rather than classify is novel.
- Very detailed theoretical analysis is provided which makes the result be persuasive.
- Experiments are well thought out and highlight the parameter study which makes the results of the work reproducible.

Cons:
- Lemma 4.1 “Optimal Discriminator for GRDA”. This lemma is intuitively correct. However, the authors don’t show the strict analysis to this basic assumption, that is, the optimal discriminator will output the conditional expectation of A_ij over all possible combinations of domain pairs (i, j) sampled from p(u|e) and p(u|e’). Maybe giving an example would help readers understand this better.
- Proposition 4.1 “Chain of Three Nodes”. This is a very interesting idea. However, there is no strict proof provided.
- The authors didn’t analyze why the change of the task of discriminator from classification to generation would help the model’s performance in domain adaptation. Some intuitive analysis would not be enough.


**Summary Of The Paper:**

This paper proposes a new model called GRDA to relax the uniform alignment which ignores topological structures among different domains. GRDA uses a domain graph to encode domain adjacency. The authors also generalize the existing adversarial learning framework with a novel graph discriminator using encoding conditioned graph embeddings. Detailed math inductions and proofs are given. Empirical results on both synthetic and real-world datasets demonstrate the superiority of this method.

**Summary Of The Review:**

Overall, being among the few papers that venture into the domain adaptation of graph, the paper introduces a number of new concepts and mechanisms. This framework is promising.

---

> ### Author Response · Authors · 2021-11-16
> **Response for your questions on theoretical analysis, etc.**
>
> Thank you for your encouraging comments. We are glad that you find our idea of “utilizing the original topological structures among different domains” interesting, our theoretical analysis persuasive, and our experiments well thought out and reproducible. Below we address your questions one by one.
>
> **Q1: Lemma 4.1 “Optimal Discriminator for GRDA”. This lemma is intuitively correct. However, the authors don’t show the strict analysis to this basic assumption, that is, the optimal discriminator will output the conditional expectation of A_ij over all possible combinations of domain pairs (i, j) sampled from p(u|e) and p(u|e’). Maybe giving an example would help readers understand this better.**
>
> Thanks for your suggestion. We did prove the Lemma 4.1 in Appendix A (all our proofs can be found here). It is worth noting that ``"optimal discriminator will output the conditional expectation of A_ij over all possible combinations of domain pairs (i, j) sampled from p(u|e) and p(u|e’)"`` is not an assumption; rather, it is the statement that we prove in Lemma 4.1. We add a pointer to the proof in the Appendix and an explanatory example in the appendix of the revised version.
>
>
> **Q2: Proposition 4.1 “Chain of Three Nodes”. This is a very interesting idea. However, there is no strict proof provided.**
>
> Thanks for mentioning this. We should have emphasized that “Chain of Three Nodes” is just a special case for “Chain Graphs”. Interestingly it can also be seen as a special case of “Star Graphs”, where 2 domains are connected to one center domain. Thus, proposition 4.1 can be derived from Corollary 4.3 or Corollary 4.4, which are both proved in Appendix A. We use this case to show that GRDA is performing relaxed domain adaptation while interpolating and extrapolating embedding distributions according to the domain graph.
>
> **Q3: The authors didn’t analyze why the change of the task of discriminator from classification to generation would help the model’s performance in domain adaptation. Some intuitive analysis would not be enough.**
>
> We have shown that our model is a generalization of the classic domain adaptation methods. When the domain graph is a clique, the GRDA recovers the uniform alignment of the classical adversarial discriminator that performs classification (Corollary 4.2). This means that our model can perform no worse than the classical model, and with the flexibility of alignment, we can achieve better performance, which is shown in Table 1, 2, and 3.
>
> According to your suggestion, we also perform additional experiments to gain more insight into GRDA. Specifically, we visualize the encodings of data produced on DG-15. All the encodings are reduced to 2 dimensions by TSNE.
>
> Figure 9(a) in the Appendix shows the domain graph for DG-15. To showcase GRDA’s ability to align encodings according to the domain graph, Figure 8(a) in the Appendix plots the encodings of domains 0, 1, 7, and 9 produced by GRDA. We can see the encodings from adjacent domains (i.e., domains 9, 7, and 1) align with each other, while the encodings from distant domains (i.e., domains 0 and 9) do not align with each other. This demonstrates that our model successfully relaxes uniform alignment according to the domain graph.
>
> Figure 8(b)  in the Appendix shows the encodings of DANN. We can see that although domain 0 and 9 are distant, they still align with each other, showing that classical domain adaptation methods lack the flexibility of alignment compared with GRDA.

---

### Author Response · Authors · 2021-11-26
**Sincerely looking forward to further discussion**

Dear reviewers:

We thank all reviewers’ effort in evaluating this work. According to rNQp’s suggestion, we have added a new state-of-the-art entropy-based method, SENTRY (Prabhu et al., 2021) from ICCV 2021 as a baseline. We totally understand that you might be busy with other engagements, but we still very much look forward to any constructive feedback for our rebuttal.

The following table shows the results for SENTRY on two datasets, DG-15 and DG-60. We can see that while SENTRY is better than Source-Only and some other baselines such as ADDA (on DG-60), our proposed GRDA can still outperform SENTRY by a large margin.

Note that entropy-based methods like SENTRY only work for classification tasks; therefore it is not applicable to TPT-48, our real-world regression dataset. We are currently running experiments on CompCar, our real-world classification dataset, and will post the results once they are available.

| Method | Source-Only | DANN | ADDA | CDANN | MDD | SENTRY | GRDA (Ours) |
| :----: | :----: |:----: |:----: |:----: |:----: | :----: | :----: |
| DG-15 | 39.77 | 72.11 | 69.00 | 72.44 | 51.33 | 43.67 | 84.44 |
| DG-60 | 38.39 | 61.98 | 32.17 | 61.70 | 66.24 | 47.80 | 96.76 |
| | | | | | |


Despite the best of efforts, confusion may still exist with only a single round of interaction. With this new experiment result, we warmly welcome any additional questions or discussions to address any remaining concerns or potential misunderstanding. We believe that the reviewer, the AC and us can reach a consensus on our work at the end, and we sincerely hope to hear from you soon.

Thanks,

GRDA authors


Viraj Prabhu, Shivam Khare, Deeksha Kartik, and Judy Hoffman.  Sentry: Selective entropy optimization via committee consistency for unsupervised domain adaptation. In Proceedings of theIEEE/CVF International Conference on Computer Vision, pp. 8558–8567, 2021

---

### Decision · Program_Chairs · 2022-01-20

**Decision:**

Accept (Poster)

**Comment:**

This paper proposes to leverage topological structure between domains, expressed as a graph, towards solving the domain adaptation problem.

Reviewer n4Lk thought the ideas were interesting, appreciated the theoretical analysis and indicated that the experiments were “well thought out”. The reviewer asked for more detail on Lemma 4.1 and suggested that a proof be provided for Proposition 4.1. They asked for more justification on why the change of task for the discriminator from classification to generation would improve performance. The authors responded to these comments, clarifying the proof of Lemma 4.1 in the appendix. They clarified that proposition 4.1 can be derived from Corollary 4.3 or Corollary 4.4. On the point of classical vs. enhanced discriminator the authors provided additional experiments.

Reviewer rNQp commented that the method was easy to follow and noted the theoretical and empirical analysis. They expressed some concern that previous work on graph-based domain adaptation was inadequately addressed. Like reviewer n4Lk they seemed unconvinced that the proposed graph discriminator was an improvement over past SOTA and questioned its novelty. In terms of claims about novelty and competitiveness relative to previous works I would have liked to see the reviewer make specific references rather than criticize in general terms. The authors’ responded to the reviewer, adding a recent entropy-based method (SENTRY) to the experiments and showed that their method outperformed this ICCV 2021 work by a large margin. They responded to the reviewer’s remarks about the original discriminator and variants, pointing out that this was already established in the paper. They used the other reviews to dispute the claim of lack of novelty.

Reviewer uDYW felt that the work was novel and interesting. Like rNQp they thought the paper was clear. They questioned the practical advantage over baselines. The authors responded to the reviewer’s question about using a data graph. They responded to the question about parameter tuning and computational cost. They addressed the question about limited improvements in real-world datasets.

I had some difficulty motivating the reviewers to engage in the discussion and acknowledge the authors’ response. The authors also politely attempted to nudge the reviewers to consider their updated results. In my opinion, the author responses have addressed most of the reviewer concerns  and I don’t see any critical issues remaining. Therefore I think that this paper should be accepted as a poster.